# The E3 ubiquitin ligase ZNRF2 is a substrate of mTORC1 and regulates its activation by amino acids

Gerta Hoxhaj[1,2]*, Edward Caddye[1†], Ayaz Najafov[1†], Vanessa P Houde[1], Catherine Johnson[1], Kumara Dissanayake[1,3], Rachel Toth[1], David G Campbell[1], Alan R Prescott[4], Carol MacKintosh[1,3]*

[1]MRC Protein Phosphorylation Unit, School of Life Sciences, University of Dundee, Dundee, United Kingdom; [2]Department of Genetics and Complex Diseases, Harvard T.H. Chan School of Public Health, Boston, United States; [3]Cell and Developmental Biology Division, School of Life Sciences, University of Dundee, Dundee, United Kingdom; [4]Cell Signalling and Immunology Division, School of Life Sciences, University of Dundee, Dundee, United Kingdom

**Abstract** The mechanistic Target of Rapamycin complex 1 (mTORC1) senses intracellular amino acid levels through an intricate machinery, which includes the Rag GTPases, Ragulator and vacuolar ATPase (V-ATPase). The membrane-associated E3 ubiquitin ligase ZNRF2 is released into the cytosol upon its phosphorylation by Akt. In this study, we show that ZNRF2 interacts with mTOR on membranes, promoting the amino acid-stimulated translocation of mTORC1 to lysosomes and its activation in human cells. ZNRF2 also interacts with the V-ATPase and preserves lysosomal acidity. Moreover, knockdown of ZNRF2 decreases cell size and cell proliferation. Upon growth factor and amino acid stimulation, mTORC1 phosphorylates ZNRF2 on Ser145, and this phosphosite is dephosphorylated by protein phosphatase 6. Ser145 phosphorylation stimulates vesicle-to-cytosol translocation of ZNRF2 and forms a novel negative feedback on mTORC1. Our findings uncover ZNRF2 as a component of the amino acid sensing machinery that acts upstream of Rag-GTPases and the V-ATPase to activate mTORC1.

*For correspondence: ghoxhaj@hsph.harvard.edu (GH); c.mackintosh@dundee.ac.uk (CM)

†These authors contributed equally to this work

**Competing interests:** The authors declare that no competing interests exist.

## Introduction

Mechanistic target of rapamycin (mTOR) is a conserved serine/threonine protein kinase of the phosphatidylinositol 3-kinase (PI3K)-related kinase family (PIKK), which functions as the catalytic subunit of two distinct complexes, mTORC1 and mTORC2 (*Wullschleger et al., 2006*). In mTOR complex 1 (mTORC1), mTOR associates with Raptor, mLST8, PRAS40 and Deptor. The second mTOR-containing complex, mTORC2, comprises mTOR, Rictor, mSIN1, mLST8, Deptor and Protor. The functions of the two complexes can be distinguished through the use of the mTOR inhibitor, rapamycin, which acutely and specifically inhibits mTORC1, but has no short-term effect on mTORC2 (*Sarbassov et al., 2006*).

mTORC1 is a central integrator of regulatory inputs from intracellular amino acids, ATP, $O_2$ and extracellular signals such as insulin and growth factors (*Dibble and Manning, 2013*). When active, mTORC1 initiates transcriptional, translational and post-translational responses to promote a multitude of anabolic activities and to suppress catabolic processes (*Duvel et al., 2010*; *Howell et al., 2013*; *Sarbassov et al., 2005*). Through phosphorylation of its substrates, including ribosomal protein S6 kinase (S6K) and the eIF4E-binding proteins (4-EBP), mTORC1 stimulates mRNA translation and, ultimately, cell growth and proliferation (*Fingar and Blenis, 2004*; *Showkat et al., 2014*).

**eLife digest** During digestion, proteins are broken down into their constituent parts called amino acids. Amino acids are transported in the bloodstream and are used to build up new cells and repair old ones. Optimal regulation of the cellular rates of amino acid uptake and protein synthesis is critical to the overall health of our bodies.

Inside each of our cells is a molecule called mammalian target of rapamycin (mTOR for short), which acts as a controller that receives information about amino acid availability. mTOR also senses how much of each amino acid the cell needs and calibrates the cell's amino acid uptake and protein synthesis machineries accordingly.

When investigating an enzyme named ZNRF2, Hoxhaj et al. discovered that it interacts with mTOR on membranes inside cells. This raised questions about how ZNRF2 might work with mTOR to sense amino acid supplies and regulate cell growth.

Hoxhaj et al. found that when cells are provided with amino acids and growth-stimulating hormones, mTOR is activated and attaches a phosphate group to ZNRF2. This chemical modification promotes the release of ZNRF2 from membranes so that ZNRF2 separates from mTOR. In contrast, when cells are starved of amino acids, this phosphate group is removed from ZNRF2, which then returns to the membranes. On membranes, ZNRF2 also influences the activity of a pump called V-ATPase, which controls the internal acidity of the membrane-enclosed vesicles named lysosomes that help to recycle amino acids inside cells. The action of ZNRF2 on the pump may help to prime mTOR so that it is ready to sense amino acids.

These findings by Hoxhaj et al. suggest that ZNRF2 and mTOR may 'tune' each other, making constant to-and-fro adjustments to help ensure that levels of amino acid uptake and cell growth are set just right. However, many questions about ZNRF2 still remain to be addressed. For example, are genetic mutations in ZNRF2 involved in cancers, developmental disorders or growth syndromes? Is ZNRF2 most important in the brain, where it is particularly abundant? And how does ZNRF2 affect acidity within the lysosomes?

mTORC1 activation involves its recruitment onto late endosomes and lysosomes (*Ogmundsdottir et al., 2012*; *Sancak et al., 2010*; *Zoncu et al., 2011*). Growth stimuli regulate mTORC1 via the heterotrimeric TSC1-TSC2-TBC1D7 complex, which negatively regulates the Rheb-GTPase, an activator of mTORC1 (*Dibble and Manning, 2013*). In parallel, amino acids signal to mTORC1 via the Rag GTPases, which consist of RagA or RagB bound to RagC or RagD (*Hirose et al., 1998*; *Kim et al., 2008*; *Sancak et al., 2008*). Amino acids, through the Rag-GTPases, promote mTORC1 recruitment and activation at the lysosomal surface, where Rheb resides (*Dibble and Manning, 2013*; *Menon et al., 2014*; *Sancak et al., 2008*). Rag-GTPases are anchored to the lysosomes via the Ragulator complex that comprises of p18, p14, MP1, HBXIP and C7orf59 proteins (*Bar-Peled et al., 2012*; *Sancak et al., 2010*). This pentameric Ragulator complex was shown to possess guanine nucleotide exchange factor (GEF) activity towards RagA and RagB (*Bar-Peled et al., 2012*). More recently, work from the Sabatini group identified the GATOR1 complex as a GTPase-activating protein (GAP) for the RagA/B GTPases, and as a major negative regulator of the amino acid sensing pathway, loss of which causes mTORC1 signalling to be completely insensitive to amino acid deprivation (*Bar-Peled et al., 2013*).

Interestingly, the vacuolar proton-ATPase (V-ATPase) was reported to be involved in activation of mTORC1 by amino acids (*Zoncu et al., 2011*), however the mechanisms linking the V-ATPase, amino acids and mTORC1 activation are still unclear. The V-ATPase pump was shown to interact with the Ragulator complex in an amino acid-sensitive manner. This pump also supports the amino acid-promoted interaction between Ragulator and Rags, and the export of amino acids from the lysosome *via* PAT1 (proton-assisted transporter 1) (*Bar-Peled and Sabatini, 2014*; *Bar-Peled et al., 2012*; *Nada et al., 2014*; *Zoncu et al., 2011*). The V-ATPase is a large, multisubunit $H^+$ pump composed of V1 (catalytic) and V0 (membrane-spanning) subcomplexes. At the surface of membrane vesicles, V-ATPase couples the energy of ATP hydrolysis to proton translocation across plasma and intracellular membranes, which results in acidification of intracellular compartments such as secretory vesicles,

early and late endosomes and lysosomes (*Forgac, 2007*). Inhibition of the V-ATPase by compounds such as conconamycin A or bafilomycin A results in increased lysosomal pH, as well as inhibition of the mTORC1 (*Hinton et al., 2009*).

Our previous study showed that the E3 ubiquitin ligase ZNRF2 is an enzyme tethered to intracellular membranes, via an N-myristoyl moiety, where it ubiquitylates the Na$^+$/K$^+$ATPase pump (*Hoxhaj et al., 2012*). ZNRF2 is robustly phosphorylated on Ser19, Ser82 and Ser145 in response to growth factors, phorbol ester (PMA) and forskolin. Akt and PKC were identified as kinases phosphorylating of Ser19 and Ser82, respectively, and these sites are responsible for mediating the binding of ZNRF2 to 14-3-3 proteins (*Hoxhaj et al., 2012*). Furthermore, the phosphorylations of Ser19 and Ser145 promote the release of ZNRF2 from intracellular membranes into the cytosol in an Akt-dependent manner (*Hoxhaj et al., 2012*).

Here, we show that ZNRF2 is a regulator of mTORC1 activation by amino acids. Upon growth factor and amino acid stimulation, mTORC1 phosphorylates ZNRF2 at Ser145 promoting its dissociation from membranes. We also show that the protein phosphatase 6 (PP6) dephosphorylates ZNRF2 at Ser145, re-localizing ZNRF2 to the membranes. Interestingly, we also find that on membranes ZNRF2 interacts with the V-ATPase and positively regulates its functions. Our findings present ZNRF2 as a positive regulator of nutrient-mediated mTORC1 signalling, which is also a negative feedback target of mTORC1 signalling.

## Results

### ZNRF2 interacts with mTOR

To better understand the molecular function of ZNRF2, we aimed to identify ZNRF2-interacting proteins. To do this, extracts of HEK293 cells stably expressing GFP-ZNRF2 (N-terminal tag, non-myristoylated) and ZNRF2-GFP (C-terminal tag, myristoylated) were subjected to immunoprecipitation. After SDS-PAGE, strong bands at the molecular weights expected for the GFP-tagged ZNRF2 proteins were identified as such by mass spectrometric analyses (*Figure 1—figure supplement 1a,b*). As reported previously, the E2 conjugating enzyme UBE2N/UBC13 co-purified with both forms of ZNRF2, whereas the Na$^+$/K$^+$ATPase ATP1A1 subunit co-purified only with the N-myristoylated ZNRF2-GFP protein (*Hoxhaj et al., 2012*). In addition, we identified mTOR as a high-score hit in the immunoprecipitates of N-myristoylated ZNRF2-GFP protein (*Figure 1—figure supplement 1b*). The interactions of mTOR with ZNRF2 was confirmed by Western blotting, which showed that endogenous mTOR bind to ZNRF2-GFP, but not to the GFP-only control nor to an N-myristoylation-defective mutant (G2A) of ZNRF2 (*Hoxhaj et al., 2012*), indicating that N-myristoylation of ZNRF2 is important for this interaction (*Figure 1a*). ZNRF2 also interacted with other components of the mTORC1 complex, namely raptor and mLST8 (*Figure 1b* and *Figure 1—figure supplement 1c*) and showed co-localization with mTOR in HEK293 cells (*Figure 1—figure supplement 1d*). To test whether the binding of ZNRF2 to mTOR was direct or mediated by one of the mTORC1 or mTORC2 components, we immunoprecipitated ZNRF2-GFP from cells depleted of Raptor or from mouse embryonic fibroblast (MEF) cells lacking rictor, Sin1 and mLST8 (*Figure 1—figure supplement 1e,f*, respectively). ZNRF2 interacted with mTOR under all these conditions, indicating that raptor, rictor, Sin1 and mLST8 do not mediate the binding of ZNRF2 to mTOR (*Figure 1—figure supplement 1e, f*). We next aimed to identify regions in ZNRF2 responsible for interacting with mTOR. mTOR bound to both the N-terminal (1 to 156) region of ZNRF2 and C-terminal RING (catalytic) domain-containing region of ZNRF2, provided these fragments also carried an N-terminal myristoyl group (*Figure 1c*), but was not able to interact to ZNRF2 with a defective UBZ domain (Cys160Ala/Cys163Ala mutations) or to the RING domain alone (residues 156-end). These data indicate that membranal localization is required for the mTOR-ZNRF2 interaction, consistent with lack of mTOR binding to non-N-myristoylated ZNRF2 (*Figure 1a*), and suggest that while the UBZ domain is involved, more than one domain of ZNRF2 contributes to mTOR binding.

### ZNRF2 is a substrate of mTORC1

We hypothesized that ZNRF2 is a substrate of mTOR. To explore this, we first performed in vitro kinase assay using bacterially purified ZNRF2-GST as a substrate of mTOR, which was immunoprecipitated from HEK293 cells. mTOR could phosphorylate ZNRF2-GST in vitro, and this was prevented

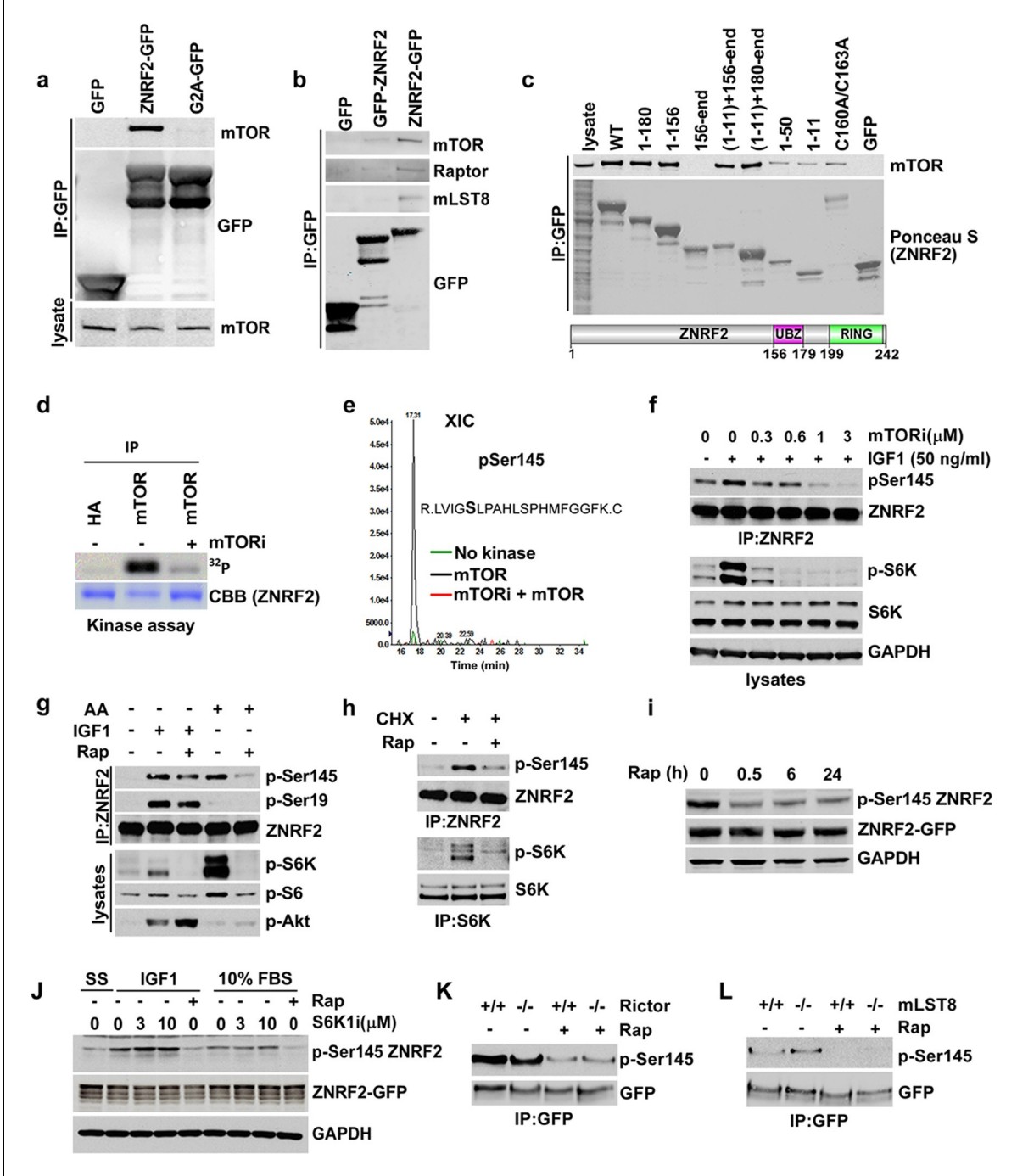

**Figure 1.** ZNRF2 binds to mTOR and is a substrate of mTORC1. (**a**) Lysates of cells expressing GFP control, ZNRF2-GFP and a myristoylation mutant G2A-GFP (with GFP-tag at the C-terminus) were subject to immunoprecipitation with GFP-Trap beads. Precipitates were immunoblotted with the indicated antibodies. (**b**) HEK293 Flp-In cells that stably express GFP or GFP-tagged (N-terminus and C-terminus) ZNRF2 were harvested in 0.3% CHAPS lysis buffer and GFP tagged proteins were immunoprecipitated with GFP-Trap beads. Precipitates were analyzed by Western blotting with the indicated antibodies. (**c**) Full-length ZNRF2 and the indicated fragments of ZNRF2 (all with C-terminal GFP tags) were tested for binding to endogenous mTOR. C160A/C163A represents ZNRF2 with the zinc-coordinating cysteines of the UBZ domain changed to alanines. Below, schematic diagram of full-length ZNRF2 containing UBZ (zinc finger) and Ring (catalytic) domains is shown. (**d**) Kinase assays were performed using mTOR immunopurified from HEK293 cells and bacterially purified GST-ZNRF2 as substrate in the presence of $Mg^{2+}[\gamma\text{-}^{32}P]ATP$ for 30 min. Where indicated, 1 μM of catalytic mTOR inhibitor (Ku-0063794) was added to the reaction and incubated for 10 min at 4°C before the addition of the GST-ZNRF2. Reactions were subjected to SDS-PAGE and autoradiography. CBB, Coomassie brilliant blue. (**e**) As in (**d**), except that phosphorylated GST-ZNRF2 was digested with trypsin and tryptic peptides were analyzed by LC-MS on an ABI 4000 Q-TRAP system using precursor ion scanning in negative mode, searching for the $(PO_3^-)$

*Figure 1 continued on next page*

*Figure 1 continued*

ion (-79 Da). The extracted ion chromatograph for phospho-Ser145 is presented. (f) HEK293 cells stimulated with IGF1, in the presence or absence of the indicated amounts of catalytic mTOR inhibitor (Ku-0063794), were tested for phosphorylation of S145 by Western blotting after immunoprecipitating endogenous ZNRF2. The phosphorylation status of Thr389 p70S6-kinase (p-S6K) was assayed. p-S6K is a marker of mTORC1 activation. S6K and GAPDH were used as controls. (g) HEK293 cells were starved of amino acids (1.5hr) and stimulated with IGF1 or amino acids, in the presence or absence of rapamycin. Endogenous ZNRF2 was immunoprecipitated and its phosphorylation (p-S19 and p-S145) was assayed by Western blotting. (h) HEK293 cells were starved of amino acids as in (g) and cycloheximide was added in the presence or absence of rapamycin. Phosphorylation of ZNRF2 was assayed as in (g). (i) Cell lysates of HEK293 Flp-In cells stably expressing ZNRF2-GFP were treated with 20 nM rapamycin for the indicated times. Phosphorylation of ZNRF2 was assayed as in (g). (j) HEK293 Flp-In cells that stably express ZNRF2-GFP were stimulated with IGF1 or serum in the presence or absence of the indicated amounts of S6K1 inhibitor (PF-4708671). Cell lysates were tested for phosphorylation of S145 of ZNRF2. (k–l) Rictor and mLST8 wild-type and knock-out MEFs were transfected with ZNRF2-GFP for 36 hr. Cells were treated with 20 nM rapamycin for 30 min in fresh media containing 10% FBS. ZNRF2-GFP was immunoprecipitated and precipitates were immunoblotted with p-S145 antibody of ZNRF2.

The following figure supplements are available for figure 1:

**Figure supplement 1.** ZNRF2 is a target of mTORC1 and interacts with mTOR.

**Figure supplement 2.** Conservation of ZNRF2 in eukaryotes.

**Figure supplement 3.** Ser145 phosphorylation of ZNRF2 promotes its dissociation from membranes.

by the mTOR-specific inhibitor Ku-0063794 (*Figure 1d*). Mass spectrometric analysis revealed a single phosphorylated residue, namely Ser145 (*Figure 1e*). Though ZNRF2 is conserved across eukaryotes (*Figure 1—figure supplement 2a–c*), Ser145 is conserved only in the vertebrate proteins (*Figure 1—figure supplement 2c*). To validate this phosphorylation, we raised a phospho-specific antibody against Ser145. In cells, the mTOR catalytic inhibitor, Ku-0063794, decreased Ser145 phosphorylation in response to IGF1 (*Figure 1f*). Since this compound inhibits both mTORC1 and mTORC2, to distinguish between these complexes, we selectively stimulated mTORC1 with either amino acids or cycloheximide (CHX) in the absence of growth factors (*Figure 1g,h*, respectively). Ser145 was markedly phosphorylated in response to these treatments and this effect was blocked by the mTORC1 specific inhibitor, rapamycin (*Figure 1g,h*), suggesting that mTORC1 rather than mTORC2 mediates Ser145 phosphorylation. However, while mTORC1 is solely responsible for amino acid-stimulated phosphorylation of Ser145, rapamycin decreased, but did not abolish Ser145 phosphorylation in response to IGF1 (*Figures 1g*), which suggests that another IGF1-activated kinase can also phosphorylate this site. The phosphorylation of Ser145 was not affected by the p70 S6 kinase 1 (S6K1)-specific inhibitor PF-4708671 (*Pearce et al., 2010*), indicating that ZNRF2 is directly phosphorylated by mTORC1 rather than via S6K (*Figure 1j*). Moreover, phosphorylation of Ser145 was not decreased in rictor or mLST8 knock-out MEFs, ruling out the possibility that mTORC2 phosphorylates this site (*Figures 1k and 1l*, respectively). Together, these data are consistent with Ser145 of ZNRF2 being phosphorylated by mTORC1 in response to growth factors and amino acids.

Previously, we reported that ZNRF2 is translocated from intracellular membranes to the cytosol upon IGF1 stimulation and that Ser145Ala mutation hampered the IGF1-stimulated dissociation of ZNRF2 into the cytoplasm (*Hoxhaj et al., 2012*). Consistent with mTORC1 being a Ser145 kinase, rapamycin increased the membranal localization of ZNRF2-GFP (*Figure 1—figure supplement 3a*; *Hoxhaj et al., 2012*). In addition, Ser145Ala mutation of ZNRF2 resulted in a more distinct membranal localization in comparison to the wild-type protein (*Figure 1—figure supplement 3b*). The expressed ZNRF2-GFP and Ser145Ala-ZNRF2-GFP co-localized predominantly with Golgi and to a lesser extent with the lysosomes (*Figure 1—figure supplement 3a,b*).

## PP6 interacts with ZNRF2 and dephosphorylates phosphoSer145 of ZNRF2

In addition to mTOR, several components of the protein serine/threonine phosphatase PP6 complex were identified as high-score hits in ZNRF2 immunoprecipitates analyzed by mass spectrometry (*Figure 1—figure supplement 1b*). The PP6 holoenzyme is a heteromeric complex, comprising the catalytic (PPP6C), three regulatory subunits PP6R1/2/3 (also called SIT4 phosphatase–associated protein (SAPS1/2/3) and three ankyrin repeat-domain containing regulatory subunits (ANKRD28/44/52)

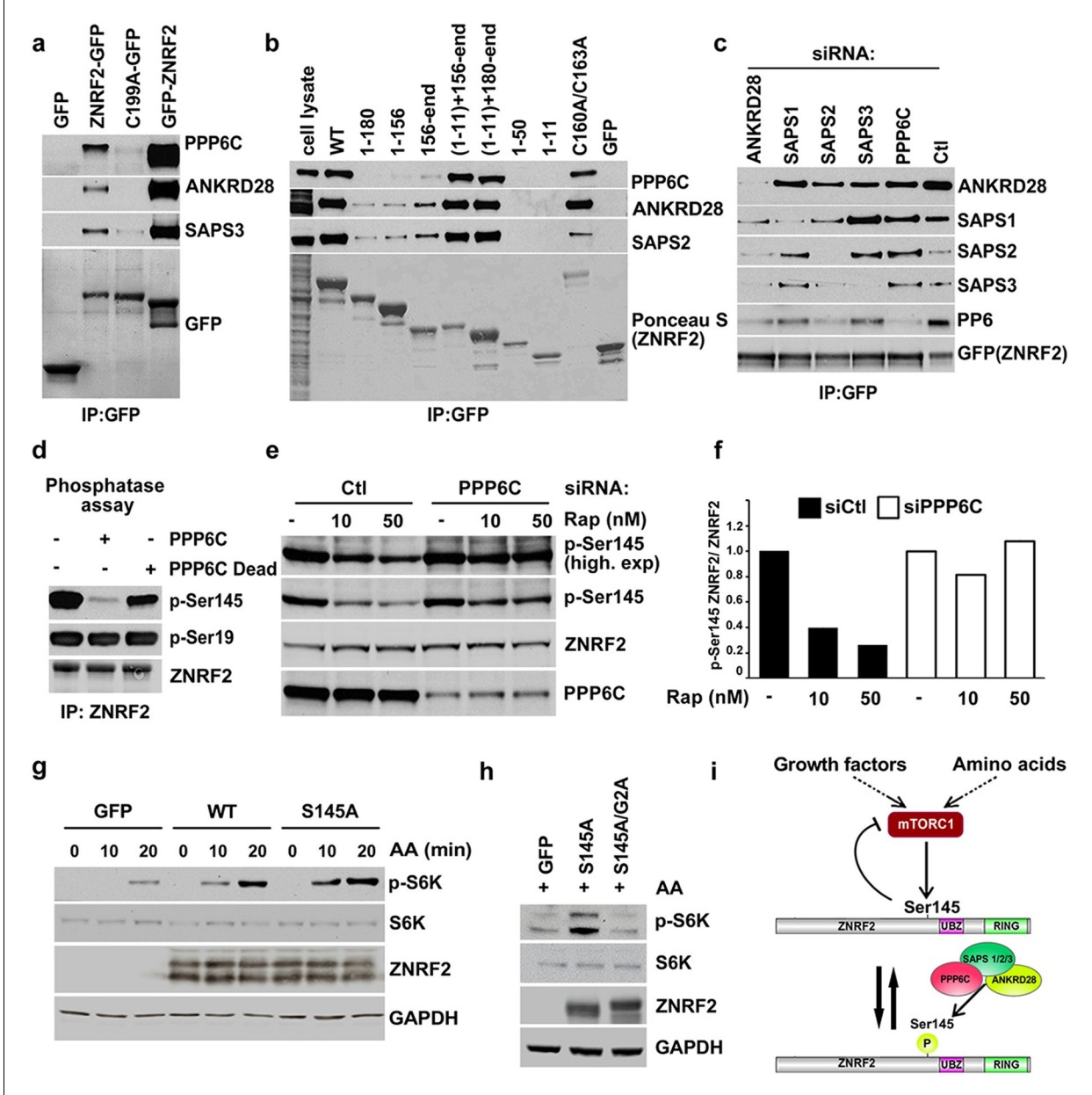

**Figure 2.** PP6 interacts with ZNRF2 and dephosphorylates phosphoS145 of ZNRF2. (a) Lysates of cells expressing GFP, GFP-ZNRF2 and ZNRF2-GFP (with GFP-tag at N- and C-terminus, respectively), and ligase-dead C199A-ZNRF2-GFP were subject to immunoprecipitation with GFP-Trap beads. Precipitates were immunoblotted with the indicated antibodies against the PP6 complex components. (b) As in *Figure 1c*, except that the immunoprecipitates were blotted for the PP6 complex component. (c) HEK293 Flp-In cells stably expressing ZNRF2-GFP were transfected with siRNAs targeting individual PP6 components or control (Ctl). Cell extracts were subjected to immunoprecipitation with GFP-Trap and analyzed by Western blotting with the antibodies indicated. (d) Purified ZNRF2-GFP was incubated with wild-type or catalytically-inactive (H55Q/R85A) recombinant PPP6C-FLAG for 2 hr and then analyzed for phosphorylation of p-S145 and p-S19 of ZNRF2. (e) HEK293 Flp-In cells expressing ZNRF2-GFP were transfected with control (Ctl) siRNA or siRNA targeting PPP6C and were incubated with two different concentrations of rapamycin (10 and 50 nM) to inhibit mTORC1. p-S145 and total ZNRF2 were assessed. (f) The ratio of p-S145 ZNRF2/ZNRF2 of *Figure 2e* is presented. (g) HEK293 cells were transfected to express GFP (control) and untagged ZNRF2 (wild-type and S145A mutant) were starved of amino acids (1.5hr), and stimulated with amino acids for 10 or 20 min. Western blotting was used to measure the phosphorylation status of S6K and levels of S6K and ZNRF2. GAPDH was used as the loading control. (h) HEK293 transfected with GFP control or S145A mutant or a combination of S145A with myristoylation mutant G2A were starved of amino acids and stimulated with amino acids for 20 min. Cell lysates were immunoblotted as in (g). (i) Model depicting ZNRF2 which is phosphorylated on Ser145 by mTORC1 and this site is dephosphorylated by PP6. Phosphorylation of Ser145 contributes to the release of ZNRF2 from membranes into cytosol.

*Figure 2 continued on next page*

*Figure 2 continued*

The following figure supplement is available for figure 2:

**Figure supplement 1.** PP6 interacts with and dephosphorylate ZNRF2.

(*Stefansson and Brautigan, 2006*; *Stefansson et al., 2008*). The binding of PP6 complex components to ZNRF2-GFP was unaffected by the position of the GFP tag (N- or C-terminus) suggesting that the N-myristoylation of ZNRF2 is not required for the interaction of PP6 to ZNRF2 (*Figure 2a* and *Figure 2—figure supplement 1a*). Interestingly, binding of the PP6 complex to ZNRF2 was compromised by the E3 ligase-dead (Cys199Ala) mutation (*Figure 2a* and *Figure 2—figure supplement 1a*). Consistent with this, ZNRF2 lacking the catalytic RING domain (fragments 1–180 and 1–156) did not bind to PP6 (*Figure 2b* and *Figure 2—figure supplement 1b*).

To identify which subunits mediate the interaction of the PP6 holoenzyme with ZNRF2, cells stably expressing ZNRF2-GFP were transfected with siRNAs targeting individual PP6 components (*Figure 2c* and *Figure 2—figure supplement 1c*). Only knockdown of PPP6C resulted in a decrease in the levels of other subunits in cell lysates (*Figure 2—figure supplement 1c*). Depletion of PPP6C, PP6R1 (SAPS1) and PP6R3 (SAPS3) did not affect the ability of other PP6 holoenzyme components to interact with ZNRF2. In contrast, knockdown of PP6R2 (SAPS2) and ANKRD28 decreased the amounts of PPP6C, SAPS1 and SAPS3 that co-immunoprecipitated with ZNRF2-GFP. These data suggest that SAPS2 and ANKRD28 mediate the interaction of the PP6 complex with ZNRF2 (*Figure 2c*).

The yeast orthologue of PP6 (Sit4), functions downstream of the rapamycin-sensitive TOR complex 1 to regulate G1 to S phase cell cycle progression, Gcn2-regulated translation and expression of certain nitrogen catabolite-repressed genes (*Beck and Hall, 1999*; *Morales-Johansson et al., 2009*). PP6 is also proposed to dephosphorylate GCN2 when mTORC1 is inhibited in mammalian cells (*Wengrod et al., 2015*). Since we found that ZNRF2 is a substrate of mTORC1 and interacts with PP6, we tested whether PP6 could dephosphorylate ZNRF2 in vitro. Active PP6 holoenzyme, but not the catalytically inactive mutant, dephosphorylated phosphoSer145, but not phosphoSer19 (Akt site) in vitro (*Figure 2d*). Importantly, rapamycin treatment of HEK293 cells decreased phosphorylation of Ser145 and this was rescued by PPP6C knockdown (*Figure 2e,f*). These data suggest that Ser145 is phosphorylated by mTOR and dephosphorylated by PP6 in vivo (*Figure 2i*). We also tested the effect of PP6 knock-down on known substrates of mTORC1 such as p-S6K, p-ULK1 and 4EBP1, and observed that while PP6 knockdown increases S6K phosphorylation, it does not affect ULK1 phosphorylation or 4EBP1 mobility shift (*Figure 2—figure supplement 1d*). To our knowledge, this is the first report of PP6-dependent changes in S6K phosphorylation.

Furthermore, we investigated the role of ZNRF2 Ser145 phosphorylation on mTORC1 signaling and found that overexpression of untagged ZNRF2 significantly increased the activation of mTORC1 by amino acids, while overexpression of ZNRF2-Ser145Ala augmented this effect, suggesting that phosphorylation of Ser145 negatively regulates ZNRF2 function towards mTORC1 (*Figure 2g,i* and *Figure 2—figure supplement 1e*). This augmented activation of mTORC1 did not occur when the overexpressed ZNRF2-Ser145Ala mutant also carried a Gly2Ala mutation, rendering ZNRF2 non-myristoylatable (*Figure 2h*). Interestingly, enhancement of mTORC1 activation by amino acids upon expression of wild-type or Ser145A ZNRF2 was decreased 2-fold by ligase dead C199A mutation (*Figure 2—figure supplement 1f*). These findings suggest that both membranal localization of ZNRF2 and its E3 ubiquitin ligase activity are important for its effect on mTORC1 activation by amino acids.

## Knockdown of ZNRF2 decreases mTORC1 activation by amino acids

We next explored whether ZNRF2 plays a role in mTORC1 signalling. Activation of mTORC1 by amino acids was decreased in HeLa and HEK293 cells upon knockdown of ZNRF2, as detected by a decrease in phosphorylation of S6K and 4E-BP1 (*Figure 3a* and *Figure 3—figure supplement 1a–d*). Since amino acids stimulate lysosomal recruitment and activation of mTORC1 (*Sancak et al., 2008*; *2010*) we asked whether ZNRF2 affected the lysosomal translocation of mTOR. After addition of amino acids, mTOR appeared in puncta-like structures that colocalize with lysosomes. The mTOR-LAMP2 colocalization was reduced by 30% upon knockdown of ZNRF2 (*Figure 3b,c*). Moreover, ZNRF2 knockdown resulted in decreased cell proliferation and cell size in HeLa cells (*Figure 3d* and

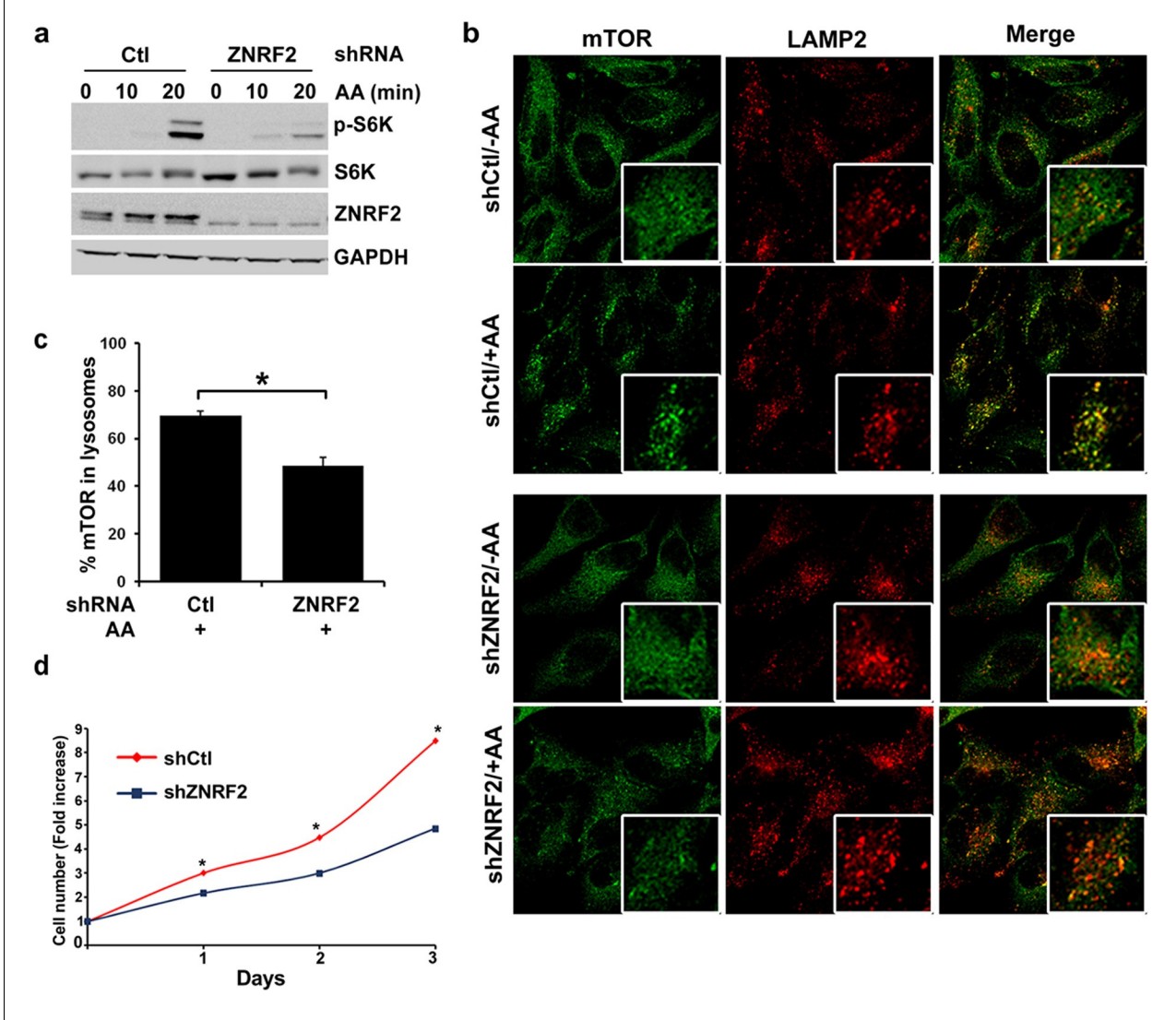

**Figure 3.** Depletion of ZNRF2 attenuates the activation of mTORC1 by amino acids . (**a**) HeLa cells expressing control shRNA (Ctl) or shRNAs targeting ZNRF2 were starved of amino acids (1.5 hr), and stimulated with amino acids for 10 or 20 min. Cell lysates were subjected to immunoblotting with the indicated antibodies. p-S6K is a marker of mTORC1 activation. (**b**) HeLa cells expressing control shRNA (Ctl) or shRNAs targeting ZNRF2 were starved of amino acids (-AA), and stimulated with amino acids for 20 min. Images of HeLa cells (with or without amino acids) co-immunostained for mTOR (green) and LAMP2 (red) are shown, together with the merged images. Inset shows a higher magnification of a selected field. (**c**) The graph displays the quantification of cells displaying lysosomal spots of mTOR fluorescence after addition of amino acids. N = ~100 cells per condition. Data are presented as mean ± S.E.M from two independent experiments. Two-tailed Student's t tests were used for the pairwise comparison. *p < 0.0001. (**d**) Cell viability of HeLa cells after knockdown of ZNRF2 was assessed daily using CellTiter-Glo Luminescent Assay. Data are mean ± S.E.M of biological triplicates from two independent experiments.

The following figure supplement is available for figure 3:

**Figure supplement 1.** Depletion of ZNRF2 decrease the activation of mTORC1 by amino acids.

*Figure 3—figure supplement 1e*, respectively). In contrast, ZNRF2 knockdown had no negative effect on activation of the PI 3-kinase pathway in response to IGF1 (*Figure 3—figure supplement 1f*). In fact, we noticed an increase in the phosphorylation of Ser473 and Thr308 of Akt and a decrease in the phosphorylation of 4E-BP1, upon ZNRF2 knockdown (*Figure 3—figure supplement 1f*), consistent with lower mTORC1 signalling relieving negative feedback via IRS1 and Grb10 (*Harrington et al., 2004*; *Hsu et al., 2011*; *Manning, 2004*; *Shah et al., 2004*; *Yu et al., 2011*).

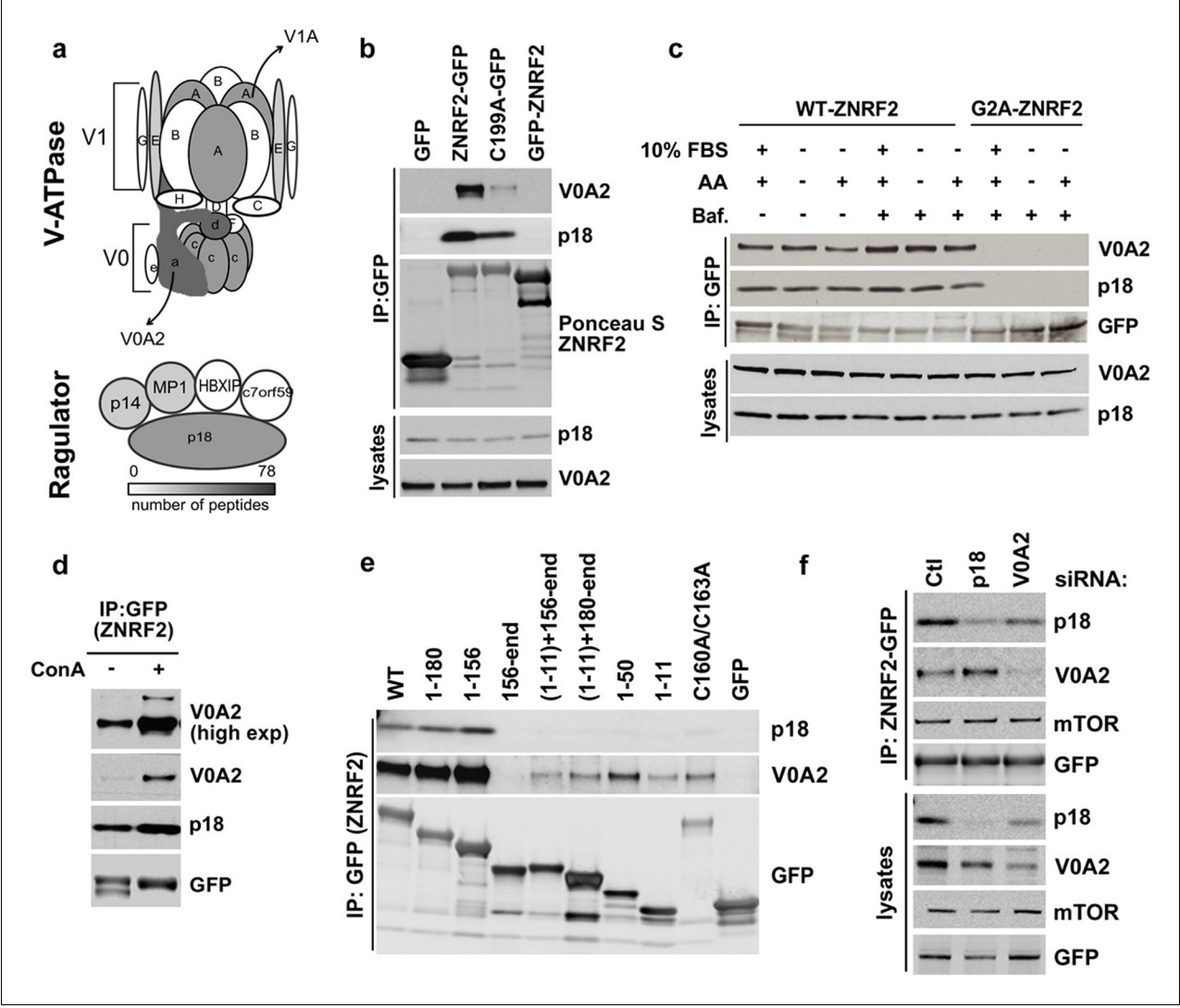

**Figure 4.** ZNRF2 interacts with V-ATPase and Ragulator. (a) Cartoon representation of V-ATPase and Ragulator subunits from mass spectrometry analyses of immunoprecipitates of ZNRF2-GFP overexpressed in HEK293 cells. The subunits are colour-coded according to the number of peptides identified by mass spectrometry and a scale is shown at the bottom. V0 and V1 subunits of the V-ATPase are presented in small or capital letters, respectively. (b) Lysates of cells expressing GFP, GFP-ZNRF2, ZNRF2-GFP (with GFP-tag at N- and C-terminus, respectively), or ligase-dead C199A-ZNRF2-GFP were subject to immunoprecipitation with GFP-Trap beads. The immunoprecipitates were immunoblotted with antibodies against the V0A2 component of V-ATPase and the p18 component of Ragulator. (c) HEK293 Flp-In cells that stably express ZNRF2-GFP or G2A-ZNRF2-GFP were treated with 2 μM bafilomycin for 60 min in the presence of serum or amino acid free medium, or amino acid free medium with 15 min stimulation with amino acids. The lysates were subjected to immunoprecipitation with GFP-Trap beads and the precipitates were blotted with V0A2 subunit of V-ATPase and p18. (d) HEK293 Flp-In cells that stably express ZNRF2-GFP were treated with 1 μM concanamycin A (ConA) for 90 min. ZNRF2–GFP was immunoprecipitated and the interaction with V0A2 subunit of V-ATPase and p18 was tested. (e) Full-length ZNRF2 and the indicated fragments of ZNRF2 (all with C-terminal GFP tags) were tested for binding to endogenous V0A2 (V-ATPase component) and p18 subunit of Ragulator. (f) HEK293 Flp-In cells stably expressing ZNRF2-GFP were transfected with siRNA targeting control (Ctl), p18 or V0A2. The lysates were subjected to immunoprecipitation with GFP-Trap beads. Precipitates were immunoblotted with the indicated antibodies.

## ZNRF2 interacts with the V-ATPase and ragulator

In the ZNRF2-GFP immunoprecipitates, we also identified components of the V-ATPase and the Ragulator complex (*Figure 4a* and *Supplementary file 1*). In particular, the V0A1/A2 components of the V-ATPase and p18 subunit of Ragulator were identified with high scores. Other V-ATPase subunits, namely V0D1, V1A, V0C and V1E; and the Mp1 and p14 subunits of Ragulator were also identified with relatively lower scores (*Figure 4a* and *Supplementary file 1*).

Similar to mTOR, V0A2 and p18 co-purified with C-terminally tagged ZNRF2-GFP, but not N-terminally tagged GFP-ZNRF2 (*Figure 4b*) nor with the Gly2Ala myristoylation-defective mutant of ZNRF2 (*Figure 4c*), indicating that myristoylation of ZNRF2 and/or its localization to membranes is required for its binding to V-ATPase and p18. The ligase-dead ZNRF2 (Cys199Ala) also displayed decreased binding to V0A2 and p18 (*Figure 4b*). Similar to mTOR, V0A2 and p18 also bind to ZNRF2 mainly through the N-terminal region of ZNRF2 (residues 1–156), although residual binding was also observed with the RING domain N-terminally fused to the N-myristoylation motif (*Figure 4e*). These experiments indicate that the binding of ZNRF2 to V-ATPase is specific and occurs at the membranes.

The binding of ZNRF2 to V0A2 was enhanced when cells were treated with bafilomycin A (Baf.) or concanamycin A (Con A), specific V-ATPase inhibitors (*Figure 4c and 4d*, respectively). However, the interactions of ZNRF2 with V-ATPase and Ragulator were not markedly regulated by amino acids (*Figure 4c*). Since p18 binds to V-ATPase, we tested whether the interaction of ZNRF2 with the V0A2 was direct or mediated by p18. Knockdown of p18 did not affect interaction of ZNRF2-GFP with V0A2, whereas knockdown of V0A2 decreased the levels of p18 in ZNRF2-GFP immuno-precipitates (*Figure 4f*). Thus, binding of ZNRF2 to V-ATPase is likely to be direct and not mediated by p18.

## ZNRF2 is important for maintaining functional lysosomes and for activation of the V-ATPase/mTORC1 axis

We next investigated whether ZNRF2 knockdown influences V-ATPase function, particularly in the context of its role in activating mTORC1 by amino acids. Known mechanisms for regulation of the V-ATPase include (*i*) reversible dissociation of the V0 and V1 subcomplexes, and (*ii*) trafficking of the V-ATPase from its site of synthesis in the endoplasmic reticulum (ER) to other membranal compartments (*Forgac, 2007*). ZNRF2 knockdown did not alter the elution profiles of the V0 and V1 subunits of V-ATPase, as assessed by size-exclusion chromatography, suggesting that ZNRF2 does not affect the association between V0 and V1 compartments (*Figure 5—figure supplement 1a*). Likewise, ZNRF2 knockdown caused no significant changes in the elution profiles of Ragulator subunit p18, mTOR, raptor or rictor (*Figure 5—figure supplement 1a*). ZNRF2 knockdown did not alter the protein levels of V-ATPase or Ragulator (*Figure 5—figure supplement 1b*). These experiments suggest that lack of ZNRF2 does not compromise the abundance or formation of V-ATPase and mTORC1 complexes.

We next assessed whether ZNRF2 is able to affect the trafficking of V-ATPase. To do this we utilized concanamycin A (ConA), a specific inhibitor of V-ATPase and Brefeldin A (BFA), a lactone antibiotic which arrests trafficking of membranal proteins from ER (*Klausner et al., 1992*). ConA binds tightly to V-ATPase and it has been proposed that *de novo* synthesis of the pump (and therefore its trafficking from ER to lysosomes) is required for the recovery from the ConA treatment (*Hanada et al., 1990*). Therefore, we knocked down ZNRF2, treated the cells with ConA or BFA, and assessed the activation of mTORC1 in the presence of amino acids after ConA or BFA were washed off. We found that the recovery of mTORC1 activity after ConA or BFA chase was markedly attenuated by ZNRF2 knockdown (*Figure 5a,b*, respectively), and enhanced by ZNRF2-Ser145Ala overexpression (*Figure 5c,d*).

Consistent with the ConA and BFA chase experiments, the knockdown of ZNRF2 resulted in decreased association of V-ATPase with lysosomes (the LAMP2 containing compartment) when V0A2 was immunopurified in the presence of a cross-linking agent (*Figure 5e*). Since one of the major functions of V-ATPase is to maintain the lysosomal pH (*Nishi and Forgac, 2002*), we tested whether knockdown of ZNRF2 affects this function of V-ATPase. In cells depleted of ZNRF2, we observed a marked increase in the lysosomal pH from 4.3 up to 5.5 in cells grown under standard conditions, and from 4.7 to 6 in amino acid-starved cells (*Figure 5f* and *Figure 5—figure supplement 1b*), indicating that ZNRF2 is crucial for the V-ATPase function.

To test whether ZNRF2 acts upstream of Rag-GTPases, we used a constitutively-active RagB^GTP-RagC^GDP complex (*Kim et al., 2008*; *Sancak et al., 2008*) that renders mTORC1 insensitive to amino acid deprivation (*Figure 5—figure supplement 1c*). Overexpression of these Rag mutants rescued the decrease in mTORC1 activity caused by ZNRF2 knockdown, suggesting that ZNRF2 acts upstream of the Rag GTPases. These experiments reinforce a model in which ZNRF2 contributes to

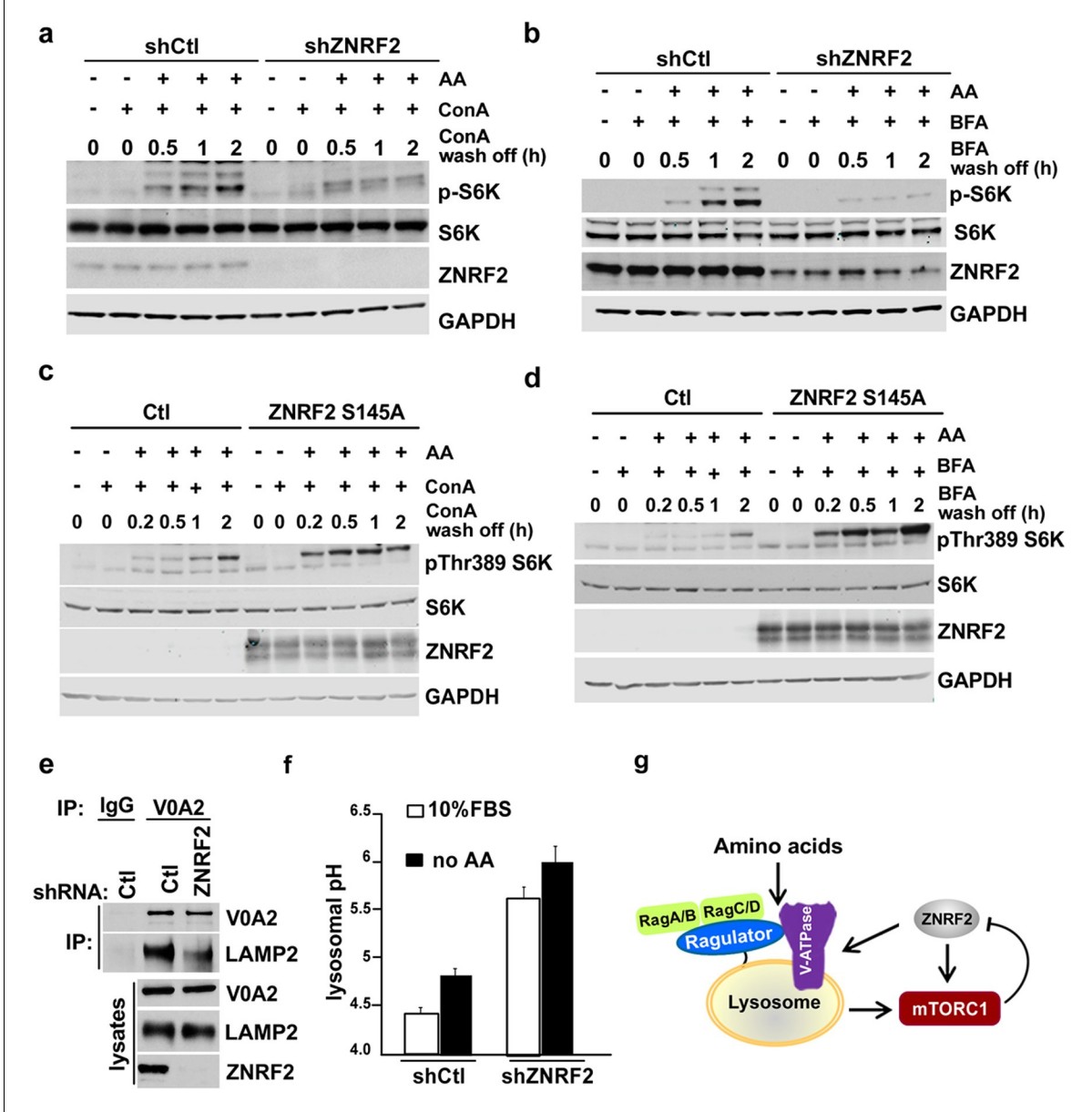

**Figure 5.** Knockdown of ZNRF2 affects V-ATPase function and lysosomal pH. (a) HEK293 cells expressing control shRNA (Ctl) or shRNAs targeting ZNRF2 were starved of amino acids in the presence of 1 µM concanamycin A (conA) for 90 min. Then, ConA containing media was washed off and cells were stimulated with amino acids for the indicated times. Cell lysates were immunoblotted with the indicated antibodies. (b) As in (a) except that cells were incubated with brefeldin A (BFA) instead of ConA. (c) HEK293 cells transfected with untagged ZNRF2 Ser145Ala or mock (Ctl) and were treated as in (a). (d) HEK293 cells transfected with untagged ZNRF2 Ser145Ala or mock (Ctl) and were treated as in (b). (e) HeLa cells expressing control shRNA (Ctl) or shRNAs targeting ZNRF2 were subjected to lysis in hypotonic buffer with chemical crosslinker DSP as described in Materials and methods. After immunoprecipitation with IgG (control) or V0A2 antibody, the LAMP2 containing compartments were assessed by immunoblotting with the indicated antibodies. (f) HEK293 cells expressing control shRNA (Ctl) or shRNAs targeting ZNRF2 were incubated for overnight with dextran-Oregon Green 514. Prior to pH measurements, the cell culture media was replaced with fresh media containing serum or amino acid-free media for 2 hr. The 490/440 fluorescence ratios were plotted as a function of pH and fitted to a Boltzmann sigmoid curve. Data are mean ± S.D. of biological triplicates and are representative of three independent experiments. (g) Model depicting the positive influence of ZNRF2 on V-ATPase, and hence amino acid activation of mTORC1. Also shown is the negative feedback in which mTORC1 phosphorylates and inhibits ZNRF2.

The following figure supplement is available for figure 5:

**Figure supplement 1.** Knockdown of ZNRF2 affects V-ATPase function and lysosomal pH.

mTORC1 activation by amino acids upstream of V-ATPase and Rag GTPases, via regulating V-ATPase localization and function (*Figure 5g*).

## Discussion

The activation of mTOR, as part of mTORC1 complex, is dependent on lysosomal localization and the vacuolar H$^+$-ATPase (V-ATPase) activity; however, the precise mechanism underlying regulation of mTORC1 by the V-ATPase remains unclear. In this study, we identified a new binding partner of mTOR, ZNRF2, which acts as a positive effector functioning upstream of V-ATPase to mediate amino acid-activation of mTORC1. Furthermore, we show that ZNRF2 is also a new substrate of mTORC1 and PP6 that promotes negative feedback regulation of mTORC1 (*Figure 2i* and *Figure 5g*).

Similar to mTOR, ZNRF2 is conserved among eukaryotes (*Figure 1—figure supplement 2*). Putative ZNRF2 orthologues in *S. cerevisiae* (Pib1p), *D. melanogaster*, *C. elegans* and vertebrates share most similarity in their UBZ and RING E3 ligase domains. The *D. melanogaster* and *C. elegans* orthologues of ZNRF2 also have a conserved N-myristoylation signal, which is critical for membranal localization of ZNRF2. Notably, the S288C strain of *S. cerevisiae* that carries a null allele for ZNRF2 orthologue Pib1p has abnormal vacuolar morphology, increased replicative lifespan and decreased growth in minimal medium, indicative of roles analogous to the V-ATPase- and mTOR-related functions of ZNRF2 that we describe in this study (http://www.yeastgenome.org).

Our data suggest that ZNRF2 is targeted via N-myristoylation to Golgi and other intracellular membranes where it interacts with the V-ATPase. According to size-exclusion chromatography analysis, knockdown of ZNRF2 has no effect on V-ATPase or mTORC1 complex assembly. However, several observations indicate that ZNRF2 promotes the trafficking of newly-synthesized V-ATPase to the lysosomes. When ER to Golgi trafficking was resumed after a BFA block and also after V-ATPase was released from inhibition by ConA, we found that ZNRF2 depletion prevented the recovery of amino acid-stimulation of mTORC1. Furthermore, ZNRF2 knockdown reduced the association of the V-ATPase V0A2 subunit with lysosomes, with a parallel increase in lysosomal pH. These data suggest that ZNRF2 is a key regulator of V-ATPase trafficking, impacting on its key function of maintaining vesicle pH. Lysosomes are acidic (pH ~4.5) intracellular organelles and their primary function is degradation of extracellular and intracellular material (*Appelqvist et al., 2013*; *Settembre et al., 2013*). Concomitantly with increased lysosomal pH, ZNRF2 depletion decreased mTORC1 signalling, consistent with requirement of functional lysosomes to sustain mTORC1 activation.

We also demonstrate that ZNRF2 knockdown affects the lysosomal recruitment of mTORC1, which is mediated by the Rag GTPases, the Ragulator complex and the V-ATPase. *Zoncu et al. (2011)* proposed an inside-out mechanism for sensing amino acids that requires the V-ATPase, which in turns interacts with the Ragulator complex. Constitutively active Rag GTPase mutants can rescue the decrease in mTORC1 activity due to knockdown of ZNRF2. This finding suggests that ZNRF2 regulates mTORC1 recruitment to lysosomes upstream of Rag GTPases. We therefore hypothesize that attenuation of mTORC1 recruitment to the lysosomal membranes upon ZNRF2 knockdown is due to impaired function of V-ATPase and subsequent effects on the Ragulator and Rag GTPases.

As well as being a positive regulator of mTORC1, ZNRF2 is also a new target of negative regulation by mTORC1, indicative of a self-regulating feedback mechanism. Our findings are consistent with ZNRF2 having been identified in an unbiased quantitative phosphoproteomics screen aiming to find mTOR substrates (*Yu et al., 2011*). Together with previous work (*Hoxhaj et al., 2012*), this study suggests that mTORC1-mediated phosphorylation of ZNRF2 on Ser145 enhances the release of ZNRF2 from intracellular membranes into the cytosol. We therefore propose the existence of an auto-inhibitory feedback loop in which the release of ZNRF2 from membranes limits the activation of mTORC1 in response to amino acids and growth factors, perhaps preventing unrestrained cell growth. Consistent with this proposal, overexpressing wild-type ZNRF2 enhanced cellular mTORC1 activation by amino acids, and expressing a Ser145Ala mutant did so to greater extent. This finely balanced dynamic regulation of ZNRF2 is reminiscent of the negative feedbacks by mTORC1 via S6K on IRS1 (*Harrington et al., 2004*; *Shah et al., 2004*) and Grb10 (*Hsu et al., 2011*; *Yu et al., 2011*).

In our model, mTORC1 mediate release of ZNRF2 from membranes into the cytosol, resulting in decreased mTORC1 activity through lysosome basification via the V-ATPase pathway. Hampering activation of mTORC1 by amino acids, during strong activation of the pathway by growth factors/insulin, may ensure that hyperactivation of mTORC1 does not occur. Consistent with this proposal,

ZNRF2 knockdown results in decreased mTORC1 signalling in cells lacking TSC2, that have constitutively active mTORC1 (data not shown). In other words, ZNRF2 adjusts the homeostatic set point for mTORC1.

We also show that the mTORC1-phosphorylated Ser145 on ZNRF2 is dephosphorylated by PP6, making PP6 a positive regulator of ZNRF2 by promoting its rebinding to membranes where it can regulate the V-ATPase and mTORC1.PhosphoSer145 of ZNRF2 should provide a useful reporter for studying how PP6 is regulated, which is critical because the relative activity of PP6 versus mTORC1 will set the level of membrane bound ZNRF2, mTORC1 activity and cell growth.

In summary, this study introduces ZNRF2 as a positive regulator of mTORC1 activation by amino acids, which functions upstream of the V-ATPase and of Rag-GTPases. ZNRF2 is also identified as a negative feedback target of mTORC1 signalling.

## Materials and methods

### Antibodies and reagents

Antibodies to mTOR, rictor, raptor, mLST8, p18, phosphoThr1135 rictor, phospho-Ser240/244 S6 ribosomal protein, total S6 ribosomal protein, phospho-Thr389 p70 S6 kinase 1, phospho-Ser65 and phospho-Thr37/46 4E-BP1, p-Ser757 ULK1, phospho-Ser473 Akt, phospho-Thr308 Akt and GAPDH were from Cell Signaling Technology. Monoclonal anti-FLAGM2-Peroxidase (HRP) antibody, Flag M2 antibody, anti-FLAGM2 Affinity Gel, anti-HA and dimethyl pimelimidate were from Sigma Aldrich; PP6 subunit antibodies (SAPS1, SAPS2, SAPS3 and ANKRD28) from Bethyl laboratories; antibodies to ATP6V1B2, ATP6V0A2 and LAMP2 from Abcam; antibody to ATP6V1A was from GeneTex; polyethylenimine (PEI) from Polysciences; Protein G-Sepharose, glutathione-Sepharose and enhanced chemiluminescence Western blotting kit were from Amersham Bioscience; [$^{32}$P-γ]ATP was from Perkin Elmer; protein G-Sepharose and immobilized glutathione from Pierce; Precast NuPAGE polyacrylamide Bis-Tris gels, Colloidal Coomassie, LDS sample buffer and Dextran Oregon Green 514 (D-7176) were from Invitrogen; dialyzed Fetal Bovine Serum (Cat # 26400036) was from Thermo-Fisher Scientific; sequencing-grade trypsin and CellTiter-Glo was from Promega; and microcystin-LR was purchased from Dr Linda Lawton (Robert Gordon University, Aberdeen, UK).

### Antibodies used for immunofluorescence studies

mTOR antibody was from Cell Signaling Technology (#2983) and LAMP2 antibody from Abcam #ab25631. Secondary antibodies were from Life Technologies (Alexa Fluor 488 Donkey Anti-Rabbit IgG and Alexa Fluor 594 Donkey Anti-Mouse IgG).

### Cell culture

Cells were maintained in Dulbecco's Modified Eagle's Medium (DMEM) with 10% foetal bovine serum (FBS). For amino acid-stimulation experiments, cells were incubated in amino acid-free EBSS for 1.5 hr, followed by addition of amino acids (50X stock of GIBCO MEM Amino Acids Solution) to 1X final concentration for 20 min, unless otherwise noted. For IGF1 stimulation, cells were serum starved in DMEM (12 hr) and stimulated with 50 ng/ml for 20 min.

### Cell lines

The following wild-type and knock-out mouse embryonic fibroblasts (MEFs) were obtained from other institutions: mLST8 MEFs (*Guertin et al., 2006*) from David Sabatini (Whitehead Institute for Biomedical Research, USA), Rictor MEFs (*Shiota et al., 2006*) provided by Manus Magnuson (Vanderbilt University School of Medicine, USA) and Sin1 MEFs (*Jacinto et al., 2006*) from Bing Su (Yale School of Medicine, USA). HEK293 and HeLa cells were purchased from ATCC and supplied by the Division of Signal Transduction Therapy (DSTT), University of Dundee. Flp-In T-REx 293 stable cell lines with tetracycline-inducible wild type or mutant forms of ZNRF2 have been described previously (*Hoxhaj et al., 2012*). All cells described above were regularly tested for mycoplasma contamination.

## Immunoblotting

After treatments, cells were rinsed with ice-cold PBS and lysed in ice-cold Triton X-100 lysis buffer. Protein concentrations were normalized prior to SDS-PAGE and immunoblotting.

## Immunoprecipitation

Lysates were pre-cleared by incubating with protein G-Sepharose beads. GFP-tagged proteins were isolated from 2 to 4 mg lysates using 15 µl of GFP-Trap-agarose, incubated at 4°C for 2 hr, and washed thrice with lysis buffer and eluted in 2x LDS sample buffer. For immunoprecipitations of endogenous mTOR, the mTOR antibody was covalently coupled to Protein G-Sepharose.

## In vitro kinase assay

mTOR kinase assays were performed as previously (*Sancak et al., 2007*). Briefly, endogenous mTOR or control (HA) was immunoprecipitated for 2 hr at 4°C. The kinase reactions were carried at 30°C in Hepes kinase buffer (25 mM Hepes (pH7.5), 50 mM KCl) containing 2 µg of substrate (GST-ZNRF2 or GST-p70 S6 kinase 1), 0.1 mM non-radioactive or [γ-$^{32}$P] ATP, and 10 mM $MgCl_2$ in a total volume of 50 µl.

## Immunofluorescence and microscopy of fixed cells

Cells grown on coverslips were fixed with 4% paraformaldehyde, permeabilized with 0.2% Triton X-100 for 5 min, blocked in 10% donkey serum in PBS for 30 min and incubated with primary antibody in 1% BSA/PBS overnight at 4°C. The coverslips were washed 3 times with 1% BSA/PBS and incubated with secondary antibodies (1:500). After 3 washes with 1% BSA/PBS, cells were stained with DAPI and mounted using Vectashield (Vector Laboratories, CA, USA). The slides were viewed under a Zeiss LSM700 microscope using an alpha Plan-Apochromat ×100 NA (numerical aperture) 1.46 objective.

## Purification of ZNRF2 and mass spectrometric analysis

HEK293 Flp-In T-Rex cells (Invitrogen) stably expressing tetracycline inducible GFP-tagged (N/C-terminus) ZNRF2 or GFP were induced with tetracycline for 36 hr and lysed in Triton X-100 lysis buffer. Clarified cell lysates (200 mg) from each cell line were subjected to immunoprecipitation using of 80 µl GFP-Trap beads for 2 hr at 4°C. Beads were collected by centrifugation for 5 min at 4000 rpm and washed thrice with washing buffer 1 (0.27 M sucrose, 50 mM Tris-Cl (pH7.4), 150 mM NaCl, 1% Triton X-100, 0.1% 2-mercaptoethanol) followed by two washes with washing buffer 2 (0.27 M sucrose, 50 mM Tris-Cl (pH 7.4), 0.1% 2-mercaptoethanol). Proteins were eluted with 150 µl of 2x LDS sample buffer. Proteins (90% of each sample) were separated on SDS-polyacrylamide gels and stained with Coomassie brilliant blue. Mass spectrometric analyses were performed as in *Hoxhaj et al. (2012)*.

## Immunopurification of lysosomes

Immunopurification of lysosomes was performed as previously (*Menon et al., 2014*). HeLa cells from two 15-cm dishes (~90% confluent) were washed once with cold PBS, gently scraped into 10 ml cold fractionation buffer (140 mM KCl, 250 mM sucrose, 2 mM EGTA, 10 mM $MgCl_2$, 25 mM HEPES, pH 7.4, 5 mM glucose), pelleted by centrifugation at 400 g for 3 min at 4°C, and resuspended in 600 µl of lysis buffer containing 140 mM KCl, 250 mM sucrose, 2 mM EGTA, 10 mM $MgCl_2$, 25 mM HEPES, pH 7.4, 5 mM glucose, 1 mM orthovanadate, 1 µM microcystin, 2x protease inhibitors and 2.5 mg/ml of the cross-linking agent DSP. Cells were mechanically lysed by drawing through a 23G needle 8 times, Lysates were centrifuged at 700 g for 10 min at 4°C, yielding a post-nuclear supernatant (PNS). Normalized PNS samples of equal volume were pre-cleared for 1 hr with protein A/G-agarose beads and incubated with V0A2 antibody (Abcam) with rocking for 12 hr at 4°C and then for an additional 3 hr following addition of 20 µl of a 1:1 slurry of buffer and pre-washed protein A/G-agarose beads. Bead-immunocomplexes were washed five times in fractionation buffer.

## siRNA knock-downs

siRNA (ON-TARGETplusSMARTpool) oligos towards PPP6C, SAPS1, SAPS2, SAPS3, Raptor, TSC2 and control were from Thermo Scientific. siRNA towards p18 and ATP6V0A2 were from Sigma. Cells at 40–50% confluency were transfected using Dharmafect 1 transfection reagent following manufacturer's instructions. Briefly, for transfection of cells in a 10 cm-dish, siRNA oligos (100 nM final concentration) and Dharmafect 1 (50 µl) were incubated separately in 1 ml of Opti-MEM for 5 min, mixed gently and incubated at room temperature for a further 20 min. The mixture was added slowly to the cells and the culture media replaced with fresh after 12 hr of siRNA transfection.

## Mammalian shRNAs

The sequences of the lentiviral shRNAs targeting ZNRF2 from Sigma were:

ZNRF2#1:CCGGCCGCACATGTTTGGAGGATTTCTCGAGAAATCCTCCAAACATGTGCGGTTTTT
ZNRF2#2:CCGGGCTCGGATCTACCTTCCAGTACTCGAGTACTGGAAGGTAGATCCGAGCTTTTT

The MISSION pLKO.1-puro lentivirus plasmid vector (7 µg) containing the shRNA sequence was transfected together with packaging (7 µg) and envelope (7 µg) plasmids in HEK293T cells (T75 flasks) of 70% confluency (12 ml final) using 60 µl of PEI (1 mg/ml). The lentiviral particles were collected 72 hr after transfection, filtered (0.45 µm pore size) and used to infect HEK293 or HeLa cells in the presence of 10 µg/ml Polybrene. Lentivirus particles (5 ml/10 cm-dish) were used to infect cells at 60–70% confluency. The cells were selected with 3 µg/ml puromycin and experiments carried out within a week of infection.

## Measurement of lysosomal pH

Lysosomal pH was measured as in (*Zoncu et al., 2011*). Briefly, $10^6$ HEK293 cells were seeded in each well of a 6-well plate and treated after 8–12 hr with 30 µg/ml of Dextran-Oregon Green 514 (D-7176, Invitrogen) for 6 to 12h. Excess dye was washed out by three rinses in PBS, and cells incubated for a further 2 hr in amino acid-free EBSS or in full medium and collected via pipetting. The media was removed by centrifugation at 1000 rpm for 1 min and cells were resuspended in 200 µl physiological buffer containing 10 mM HEPES pH7.4, 2.5 mM KCl, 2 mM $CaCl_2$, 136 mM NaCl, 1.3 mM $MgCl_2$ and 5 mM glucose. The fluorescence of Dextran Oregon Green was measured in black 96-multiwell plates. The samples were excited at 440 nm and 490 nm, and data collected at 530 nm. To calculate pH, a calibration curve was created by loading HEK293 cells with Dextran-Oregon Green, washing out excess dye and resuspending the cells in $K^+$ isotonic buffers at different pH values. The calibration standards were generated using a $K^+$ solution consisting of 145 mM KCl, 10 mM glucose, 1 mM $MgCl_2$, buffered with 20 mM HEPES (for pH 7–8), 20 mM of MES (for pH 5–6.5), or 20 mM acetate (for pH 3.5–4.5). Nigericin (10 µg/ml) was also added to the calibration samples, and 490/440 ratios were fitted to a Boltzmann sigmoid using Prism and plotted as function of pH.

## Cell lysis

After treatments, the medium was aspirated and cells were rinsed once with ice-cold PBS and lysed in 0.35 ml (10-cm dish) or 0.5 ml (15-cm dish) of ice-cold Triton X-100 lysis buffer. For mTOR kinase assays CHAPS lysis buffer was used. Lysis buffer contained: 25 mM Hepes (pH 7.5), 1 mM EDTA, 1 mM EGTA, 1% Triton X-100 or 0.3% (w/v) CHAPS, 50 mM NaF, 5 mM sodium pyrophosphate, 1 mM sodium orthovanadate, 1 mM benzamidine, 0.2 mM PMSF, 0.1% 2-mercaptoethanol, 1 µM microcystin-LR, 0.27 M sucrose and one mini Complete protease inhibitor cocktail tablet (#1697498, Roche) per 10 ml of lysis buffer. Lysates were clarified, snap frozen and stored at -80°C.

## PP6 phosphatase assay

HEK293 cells were transfected to express wild-type and catalytically-inactive PPP6C-FLAG (15-cm dishes) for 36 hr and harvested in 1% Triton X-100 lysis buffer containing 50 mM Tris-HCl, pH 7.4, 150 mM NaCl, and one tablet of protease inhibitor cocktail per 50 ml. The FLAG tagged proteins were purified using FLAG beads and eluted with FLAG peptide. These eluates were incubated at 30°C with ZNRF2-GFP (purified from cells fed with serum and used as a substrate) in a 50 µl

reaction, which contained 50 mM HEPES, 100 mM NaCl, 2 mM DTT, 0.01% Brij-35 pH 7.5 and 1 mM $MnCl_2$. The proteins were analysed for phosphorylation of ZNRF2.

## Cell proliferation assays

HeLa cells expressing control shRNA (Ctl) or shRNAs targeting ZNRF2 were plated in 96 well plates. Cell viability was measured in medium containing 1% FBS using CellTiter-Glo every 24 hr for 3 constitutive days. All proliferation assays were done with biological triplicates seeded in technical quadruplicates for each condition.

## Cell size

Cells were trypsinized, washed twice in ice-cold PBS, resuspended in 1 mL PBS, and fixed with 3-ml of absolute cold-ethanol for 15 hr. Cells were washed twice with cold PBS and incubated with 1 mL of Propidium iodide solution (Sodium citrate 3,8 mM, Propidium iodide 40 μg/ml in PBS) for 45 min in the dark at 4°C prior to analysis by a Flow cytometer.

## Bioinformatics analysis

Amino acid sequence alignments, percent identity matrix and phylogeny tree compilations were performed using BLAST (http://blast.ncbi.nlm.nih.gov/Blast.cgi; blastp (protein-protein BLAST)) and Clustal Omega (http://www.ebi.ac.uk/Tools/msa/clustalo/). Amino acid sequences were retrieved from the NCBI HomoloGene database (http://www.ncbi.nlm.nih.gov/homologene).

## Acknowledgements

We thank the Division of Signal Transduction Therapy (DSTT) teams managed by James Hastie and Hilary McLauchlan at University of Dundee for protein production and antibody purification; Bob Gourlay for assisting with mass spectrometry; Axel Knebel for helping with gel filtration. We thank the UK Medical Research Council (programme grant U127084354 to C.M.), and pharmaceutical companies that support the DSTT (AstraZeneca, Boehringer-Ingelheim, GlaxoSmithKline, Merck KgaA, Janssen Pharmaceutica and Pfizer) for financial support.

## Additional information

### Funding

| Funder | Grant reference number | Author |
| --- | --- | --- |
| Medical Research Council | U127084354 | Carol MacKintosh |

The funders had no role in study design, data collection and interpretation, or the decision to submit the work for publication.

### Author contributions

GH, Conception and design, Acquisition of data, Analysis and interpretation of data, Drafting or revising the article; EC, AN, Acquisition of data, Analysis and interpretation of data, Drafting or revising the article; VPH, ARP, Acquisition of data, Analysis and interpretation of data, Contributed unpublished essential data or reagents; CJ, KD, Acquisition of data, Contributed unpublished essential data or reagents; RT, Contributed unpublished essential data or reagents; DGC, Acquisition of data, Analysis and interpretation of data; CM, Conception and design, Analysis and interpretation of data, Drafting or revising the article

### Author ORCIDs

Gerta Hoxhaj, http://orcid.org/0000-0001-6179-3583
Carol MacKintosh, http://orcid.org/0000-0001-9166-589X

## Additional files

**Supplementary files**

• Supplementary file 1. V-ATPase and Ragulator subunits identified in ZNRF2-GFP immunoprecipitates.

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
