## [Decision Letter]

Thank you for submitting your work entitled "ZNRF2 is a substrate of mTORC1 and a mediator of its activation by amino acids" for peer review at *eLife*. Your submission has been favorably evaluated by Tony Hunter as Senior Editor and three reviewers.

In this paper the authors demonstrate that ZNRF2, a membrane-associated E3 ubiquitin ligase, promotes mTORC1 activity in response to amino acids, identifying ZNRF2 as an mTOR-associated protein that interacts directly with mTOR, and showing that depletion of ZNRF2 resulted in reduced activation of mTORC1 in response to amino acids and IGF1. Mechanistically, ZNRF2 is required for proper trafficking of vacuolar ATPase (V-ATPase) to the lysosome. In ZNRF2-depleted cells, a decrease in lysosomal V-ATPase inhibits amino acid-induced mTORC1 activation. In addition, the authors showed that mTORC1 phosphorylates ZNRF2 at Ser145 and releases ZNRF2 from membranes. This ZNRF2 release from a membrane compartment inhibits amino acid-induced mTORC1 activation, serving as a negative feedback loop.

The reviewers have discussed the reviews with one another and the editor has drafted this decision to help you prepare a revised submission. The following major points will need to be addressed in a revised version.

Essential revisions:

1) The authors previously showed that Akt phosphorylates ZNRF2 at Ser19, which releases ZNRF2 from intracellular membrane compartments into the cytosol. In this study, they demonstrate that mTORC1 phosphorylates ZNRF2 at Ser145, which also releases ZNRF2 from intracellular membrane compartments (partially the lysosome) to the cytosol. Overexpression of phospho-deficient ZNRF2 (S145A and S19A) enhances membrane association of ZNRF2 (Hoxhaj et al. 2012 and Figure 1—figure supplement 2). The phospho-deficient ZNRF2 also enhances mTORC1 activation in response to amino acids. Based on these data, the authors propose a model in which the insulin-Akt pathway inhibits ZNRF2 and thus mTORC1 (Figure 5). However, this is inconsistent with previous publications showing Akt as a positive regulator of mTORC1 through inhibition of the TSC complex, a negative regulator of mTORC1. According to authors' model, growth factors would inhibit mTORC1, which is not the case even in this study (Figure 1).

2) The authors performed amino acid stimulation experiments in the absence of growth factors. Some previous publications have shown that inputs from both amino acids and growth factors, but neither alone, are necessary to activate mTORC1 in most cells (e.g., Sancack et al. Science, 2008). How do the authors explain mTORC1 activation in response to amino acids in the absence of growth factors? In this regard, it would be informative if the authors tested the effect of amino acids on mTORC1 activity in ZNRF2-depleted or overexpressed cells in the presence of growth factors.

3) Figure 2: Consistent with the authors' model that Ser145 phosphorylation is inhibitory to ZNRF2 and mTORC1, a ZNRF2-S145A mutant enhances amino acid-induced activation of mTORC1. The also authors show that protein phosphatase 6 (PP6) directly dephosphorylates ZNRF2 at Ser145. However, contrary to what one would expect given the inhibitory effect of Ser145 phosphorylation on mTORC1, phospho-S6K (an mTORC1 target) increases in PP6-knockdown cells. No explanation is provided. Also, to investigate further this apparent discrepancy, other mTORC1 targets such as ULK1 and 4E-BP should be examined.

4) The authors did not address whether ZNRF2 E3 ligase activity is required for this function, and, if so, what protein(s) it ubiquitylates. In their earlier paper, they showed that ZNRF2 isolated from HEK239 cells could ubiquitylate Na^+^/K^+^ ATPase α1 in vitro, which raises the question of whether ZNRF2 ubiquitylates a V-ATPase (or Ragulator) subunit, and whether it affects the level of any of these proteins, e.g. does ZNRF2 depletion cause a decrease in lysosomal V-ATPase abundance at steady state? While such studies may be beyond the scope of this paper, at the very least this issue needs to be discussed in the present paper; if data with a ligase-deficient mutant are available, they could be included.

5) Some other points that could be discussed but do not necessarily need to be addressed experimentally: It is not clear whether ZNRF2 is a conserved component of the mTORC1 pathway, i.e. since activation of TORC1 is a universal eukaryotic response to amino acids, how far down in evolution are ZNRF2 orthologues found, and is there any genetic evidence for a role in mTORC1 signaling in simpler organisms? Does ZNRF2 promote V-ATPase assembly or the translocation of the V-ATPase subunits from ER to the lysosome, or both? How does ZNRF2 affect mTORC1 recruitment to lysosomal membranes, which is a process dependent on the Rag GTPases? Does ZNFR2 affect Rag activity/GTP loading? Does Ser145 phosphorylation affect the interaction of ZNRF2 with mTOR?

6) The reviewers feel that the title overstates the role for ZNRF2 in amino acid sensing upstream of mTORC1, because it appears that ZNRF2 acts indirectly, and the title should be modified accordingly.

---

## [Author Response]

*Essential revisions: 1) The authors previously showed that Akt phosphorylates ZNRF2 at Ser19, which releases ZNRF2 from intracellular membrane compartments into the cytosol. In this study, they demonstrate that mTORC1 phosphorylates ZNRF2 at Ser145, which also releases ZNRF2 from intracellular membrane compartments (partially the lysosome) to the cytosol. Overexpression of phospho-deficient ZNRF2 (S145A and S19A) enhances membrane association of ZNRF2 (Hoxhaj et al. 2012 and Figure 1—figure supplement 2). The phospho-deficient ZNRF2 also enhances mTORC1 activation in response to amino acids. Based on these data, the authors propose a model in which the insulin-Akt pathway inhibits ZNRF2 and thus mTORC1 (Figure 5). However, this is inconsistent with previous publications showing Akt as a positive regulator of mTORC1 through inhibition of the TSC complex, a negative regulator of mTORC1. According to authors' model, growth factors would inhibit mTORC1, which is not the case even in this study (Figure 1).* We apologize that our previous model (Figure 5) was not an accurate portrayal of our findings and we have now amended it. We agree with the reviewer that the net effect of insulin/Akt signaling is positive regulation of mTORC1. The effect of Akt on mTORC1 activation is a sum of multiple inputs, at least two of which (phosphorylations of TSC2 and Pras40) are positive. To our knowledge, ZNRF2 is the first negative input from Akt to mTORC1. However, deconvoluting these three inputs to *prove* that Akt negatively regulates mTORC1 via ZNRF2 is not feasible (it would require generation of knock-in of TSC2-S939A/S981A/S1130A/S1132A/T1462A and Pras40-T246A mutations) and is beyond the scope of the manuscript, so we have removed it from the model.

We also amended the Discussion section accordingly. The Discussion section first considers ZNRF2 activation of mTORC1 upstream of the V-ATPase. Next, we discuss the regulation of ZNRF2 in several paragraphs that start as follows: “As well as being a positive regulator of mTORC1, ZNRF2 is also a new target of negative regulation by mTORC1, indicative of a self-regulating feedback mechanism.”

*2) The authors performed amino acid stimulation experiments in the absence of growth factors. Some previous publications have shown that inputs from both amino acids and growth factors, but neither alone, are necessary to activate mTORC1 in most cells (e.g., Sancack et al. Science, 2008). How do the authors explain mTORC1 activation in response to amino acids in the absence of growth factors? In this regard, it would be informative if the authors tested the effect of amino acids on mTORC1 activity in ZNRF2-depleted or overexpressed cells in the presence of growth factors.*

We agree with the reviewer that full activation of mTORC1 requires inputs from both amino acids and growth factors. Amino acids act through the Ragulator and Rag GTPases to localize mTORC1 to the lysosome (Kim et al., 2008; Sancak et al., 2008,2010), bringing it in close proximity to Rheb, a potent stimulator of mTORC1 kinase activity (Sancak, 2007; Menon, Dibble et al. 2014). In turn, through Akt, growth factors stimulate dissociation of the TSC complex from Rheb at the lysosomal surface, therefore leading to activation of mTORC1.

In our experimental set up, we performed amino acid starvation without growth factors for 90 minutes, prior to stimulation with amino acids. This treatment seemed to be sufficient to translocate mTORC1 away from lysosomes (Figure 3 and Figure 3—figure supplement 1), resulting in complete loss of p-S6K and p-4EBP1 (Figure 3 and Figure 3—figure supplement 1). But, during these 90 minutes, Rheb-GTP pools may not fully convert into Rheb-GDP. It is possible that under these conditions, there is sufficient active Rheb-GTP to activate mTORC1 that is potently recruited to lysosomal surface by amino acid treatment (Figure 3 and Figure 3—figure supplement 1).

We have now tested the effects of ZNRF2 knockdown on mTORC1 activation in the presence of growth factors (dialyzed FBS) and found that ZNRF2 knockdown significantly decreases mTORC1 activation by amino acids under these conditions as well (Figure 3—figure supplement 1).

*3) Figure 2: Consistent with the authors' model that Ser145 phosphorylation is inhibitory to ZNRF2 and mTORC1, a ZNRF2-S145A mutant enhances amino acid-induced activation of mTORC1. The also authors show that protein phosphatase 6 (PP6) directly dephosphorylates ZNRF2 at Ser145. However, contrary to what one would expect given the inhibitory effect of Ser145 phosphorylation on mTORC1, phospho-S6K (an mTORC1 target) increases in PP6-knockdown cells. No explanation is provided. Also, to investigate further this apparent discrepancy, other mTORC1 targets such as ULK1 and 4E-BP should be examined.* We show that PP6 dephosphorylates phosphoS145 of ZNRF2 in response to rapamycin treatment (Figure 2) and that PP6 dephosphorylates phosphoS145 in vitro (Figure 2). We further show that S145A enhances amino acid-induced mTORC1 activation suggesting that this phosphorylation induced a negative feedback to decrease mTORC1 activation by amino acids.

We agree with the reviewer that the increase in phospho-S6K (an mTORC1 target) in PP6-knockdown cells might seem counterintuitive since one would expect that Ser145 should be phosphorylated, leading to mTORC1 inhibition. However, as in the case with many other phosphatases, we think that there are very likely many other biological events affected by knockdown of PP6 (acutely or long term transcriptional regulation) which could modulate mTORC1 and therefore S6K phosphorylation independently of ZNRF2.

To further test the effect of PP6 knock-down on the activity of mTORC1 we analyzed other known substrates of mTORC1 such as p-ULK1 and 4EBP, and observed that while PP6 knockdown increases S6K phosphorylation, it does not affect ULK1 phosphorylation or 4EBP1 mobility shift (Figure 2—figure supplement 1). These observations confirm our notion that PP6 is a pleiotropic phosphatase and therefore we cannot measure a direct effect of PP6 knock-down on mTORC1 via ZNRF2 Ser145 phosphorylation increase.

*4) The authors did not address whether ZNRF2 E3 ligase activity is required for this function, and, if so, what protein(s) it ubiquitylates. In their earlier paper, they showed that ZNRF2 isolated from HEK239 cells could ubiquitylate Na^+^/*K^+^

*ATPase α1 in vitro, which raises the question of whether ZNRF2 ubiquitylates a V-ATPase (or Ragulator) subunit, and whether it affects the level of any of these proteins, e.g. does ZNRF2 depletion cause a decrease in lysosomal V-ATPase abundance at steady state? While such studies may be beyond the scope of this paper, at the very least this issue needs to be discussed in the present paper; if data with a ligase-deficient mutant are available, they could be included.* Indeed, we have previously shown that ZNRF2 can ubiquitylate and regulate Na^+^/K^+^ ATPase α1 (Hoxhaj et al., 2012). The reviewer makes a valid point, about the possibility of ZNRF2 being able to ubiquitylate V-ATPase and/or p18 component. This is a very interesting area that we most certainly would like to explore in the future but we agree, that this is out of scope for this paper. Nevertheless, we tested whether ZNRF2 depletion was able to alter the abundance of V-ATPase but we did not observe any significant changes in V-ATPase or Ragulator protein levels (Figure 5—figure supplement 1).

As the reviewers suggested, we tested the effects of a ligase-deficient mutant of ZNRF2. The enhancement of mTORC1 activation by amino acids upon overexpression of WT or S145A ZNRF2 was decreased 2-fold by ligase dead C199A mutation of WT or S145A ZNRF2 mutants (Figure 2—figure supplement 1). These data suggest that the E3 ubiquitin ligase activity of ZNRF2 is required for its effects on mTORC1 activation by amino acids.

*5) Some other points that could be discussed but do not necessarily need to be addressed experimentally: It is not clear whether ZNRF2 is a conserved component of the mTORC1 pathway, i.e. since activation of TORC1 is a universal eukaryotic response to amino acids, how far down in evolution are ZNRF2 orthologues found, and is there any genetic evidence for a role in mTORC1 signaling in simpler organisms?*

We have added information about ZNRF2 orthologues to the Discussion section and added Figure 1—figure supplement 2 to illustrate the extent of their conservation.

“Similar to mTOR, ZNRF2 is conserved among eukaryotes (Figure 1—figure supplement 2). Putative ZNRF2 orthologues in *S. cerevisiae* (Pib1p), *D. melanogaster, C. elegans* and vertebrates share most similarity in their UBZ and RING E3 ligase domains. The *D. melanogaster* and *C. elegans* orthologues of ZNRF2 also have a conserved N-myristoylation signal, which is critical for membranal localization of ZNRF2. Notably, the S288C strain of *S. cerevisiae* that carries a null allele for ZNRF2 orthologue Pib1p has abnormal vacuolar morphology, increased replicative lifespan and decreased growth in minimal medium, indicative of roles analogous to the V-ATPase- and mTOR-related functions of ZNRF2 that we describe in this study (http://www.yeastgenome.org).”

*Does ZNRF2 promote V-ATPase assembly or the translocation of the V-ATPase subunits from ER to the lysosome, or both?*

ZNRF2 does not promote V-ATPase assembly, according to size-exclusion chromatography analysis (Figure 5—figure supplement 1). However, several observations indicate that ZNRF2 promotes the trafficking of newly-synthesized V-ATPase to the lysosomes.In Figure 5 we have shown that upon knock-down of ZNRF2 the association of V-ATPase with lysosomal marker LAMP2 is decreased. Moreover, our ConA and BFA washout experiments also indicate that ZNRF2 indeed regulates trafficking of V-ATPase (Figure 5). We have amended the Discussion section accordingly.

*How does ZNRF2 affect mTORC1 recruitment to lysosomal membranes, which is a process dependent on the Rag GTPases? Does ZNFR2 affect Rag activity/GTP loading? Does Ser145 phosphorylation affect the interaction of ZNRF2 with mTOR?* We demonstrate that ZNRF2 knockdown affects the lysosomal recruitment of mTORC1 (Figure 3), which is mediated by the Rag GTPases, the Ragulator complex and the V-ATPase. Zoncu et al. (2011) proposed an inside-out mechanism for sensing amino acids which requires the V-ATPase, which in turns interacts with the Ragulator complex. It still remains unknown how the V-ATPase regulates the function of Ragulator and Rag GTPases. Our data shows that ZNRF2 binds to V-ATPase and Ragulator subunits and that ZNRF2 knockdown diminishes the association of V-ATPase with the lysosomes and affects lysosomal pH (Figure 4 and Figure 5). We hypothesize that the attenuation of mTORC1 recruitment to the lysosomal membranes could involve the impaired function of V-ATPase and its subsequent effects on the Ragulator and the Rag GTPases. It seems that ZNRF2 regulates mTORC1 recruitment to lysosomes upstream of Rag GTPases, as constitutively active Rag GTPase mutants can rescue decrease in mTORC1 activity due to knock-down of ZNRF2 (Figure 5—figure supplement 1). We have amended the Discussion section accordingly.

6) The reviewers feel that the title overstates the role for ZNRF2 in amino acid sensing upstream of mTORC1, because it appears that ZNRF2 acts indirectly, and the title should be modified accordingly.

We have modified the title to: “The E3 ubiquitin ligase ZNRF2 is a substrate of mTORC1 and regulates its activation by amino acids”.